# Sepsis and Cognitive Assessment

**DOI:** 10.3390/jcm10184269

**Published:** 2021-09-20

**Authors:** Laura C. Jones, Catherine Dion, Philip A. Efron, Catherine C. Price

**Affiliations:** 1Department of Clinical and Health Psychology, University of Florida, Gainesville, FL 32610, USA; laura.jones@phhp.ufl.edu (L.C.J.); cdion2@phhp.ufl.edu (C.D.); 2Perioperative Cognitive Anesthesia Network, Department of Anesthesia University of Florida, Gainesville, FL 32610, USA; 3Department of Surgery, University of Florida, Gainesville, FL 32610, USA; philip.efron@surgery.ufl.edu; 4Department of Anesthesiology, University of Florida, Gainesville, FL 32610, USA

**Keywords:** sepsis, cognitive assessment, aging

## Abstract

Sepsis disproportionally affects people over the age of 65, and with an exponentially increasing older population, sepsis poses additional risks for cognitive decline. This review summarizes published literature for (1) authorship qualification; (2) the type of cognitive domains most often assessed; (3) timelines for cognitive assessment; (4) the control group and analysis approach, and (5) sociodemographic reporting. Using key terms, a PubMed database review from January 2000 to January 2021 identified 3050 articles, and 234 qualified as full text reviews with 18 ultimately retained as summaries. More than half (61%) included an author with an expert in cognitive assessment. Seven (39%) relied on cognitive screening tools for assessment with the remaining using a combination of standard neuropsychological measures. Cognitive domains typically assessed were declarative memory, attention and working memory, processing speed, and executive function. Analytically, 35% reported on education, and 17% included baseline (pre-sepsis) data. Eight (44%) included a non-sepsis peer group. No study considered sex or race/diversity in the statistical model, and only five studies reported on race/ethnicity, with Caucasians making up the majority (74%). Of the articles with neuropsychological measures, researchers report acute with cognitive improvement over time for sepsis survivors. The findings suggest avenues for future study designs.

## 1. Introduction

Sepsis is one of the most common, expensive, and inadequately managed syndromes. A 2016 task force introduced an updated definition (Sepsis-3), explaining that sepsis is “a life-threatening organ dysfunction caused by a dysregulated host response to infection” [1,2,3,4]. Approximately 2 million adults are affected by sepsis within the United States annually with sepsis being responsible for one out of three hospital deaths [5,6]. Although the sepsis-associated hospitalization frequency has minimally changed for young individuals aged 18–49 years old, the hospitalization frequency has risen dramatically for those aged 65 and older, such that sepsis is currently considered a “disease of the aged” [2]. With increasing age, individuals with sepsis are more likely to endure critical illness with major organ damage resulting in chronic care conditions.

For the brain, sepsis might act as a major inflammatory stimulus and potentially increase the brain’s susceptibility to neurodegenerative disease [7,8]. Fitting with postulates of a brain reserve theory and threshold, sepsis may stimulate the deterioration of cognitive ability or enhance the risk for future progressive cognitive deterioration [9]. Brain health and cognition is also relevant to the risk of developing acute organ dysfunction [10]. By 2050, people aged 65 and older will reach 1.6 billion worldwide [11] and our healthcare system will face greater numbers of individuals with early to late-stage neurodegenerative disorders such as Alzheimer’s disease (AD) [12].

To assist sepsis cognitive research going forward, we conducted the current review to provide insight into the strengths and weaknesses of published research addressing cognition following sepsis. We summarize study designs, the inclusion of a cognitive expert on the team, if authors considered the impacts of education, sex, race/ethnicity, the timeline of testing and inclusion of a baseline, the time of testing, the type of cognitive measures, the statistical approach, and findings. We provide conclusions regarding current limitations and strengths of this literature.

## 2. Materials and Methods

The literature review process included a thorough PubMed search for publications up to 13 January 2021 (See Figure 1). Key terms included: “sepsis and cognition”, which yielded 850, “sepsis and cognitive” (850), “sepsis and memory” (699), “sepsis and thinking” (642), and “sepsis and attention” (1962). Across these search terms a total of 3050 unique articles were identified, with 2816 articles removed based on keywords missing from the article titles, and with 234 qualifying as full text reviews. Of these, 18 met all inclusion criteria. To be eligible for inclusion, articles needed to have a population with a mean age of ≥18 years, a quantitative assessment of a cognition post-septic episode, they needed to be peer-reviewed (i.e., no dissertations), and they needed to be written in English. No limits were placed on the date of publication or timeframe of the cognitive assessment.

## 3. Results

### 3.1. Author Inclusion of Neuropsychology Expert and Study Design

Of the 18 retained articles [13,14,15,16,17,18,19,20,21,22,23,24,25,26,27,28,29,30], 61% of them included a neuropsychologist as a co-author. 66.7% of studies used longitudinal designs, while 33.3% used cross-sectional designs. 

### 3.2. Age, Education, Sex, Race and Ethnicity

The mean age ranged from 49 to 80.81 years across all studies, for a grand mean of 58.30 years. Analytically, 35% of studies considered premorbid intellectual abilities such as education, with a wide educational attainment range. Three studies (16.7%) reported a 7th grade education on average, four studies (22.2%) were comprised mainly of high school graduates (12 years), and two studies (11.1%) included mainly college graduates (16 years). 

Fifteen studies (83.3%) reported on sex, which were generally evenly split across sexes with slightly more females (*n* = 1067) than males (*n* = 995).

Only five studies (27.8%) reported on race and/or ethnicity, with the majority of patients identifying as Caucasians (79.5%). Only three studies (16.7%) reported on the inclusion of patients identifying as Black/African-American, one study (5.6%) reported on individuals identifying as Hispanic, and one study (5.6%) reported on individuals identifying as Native American or other Pacific Islanders. See Table 1 for additional demographic information.

### 3.3. Baseline and Time of Testing

Few studies (17%) included baseline cognitive testing (pre-sepsis data), making it difficult to accurately identify cognitive change from premorbid abilities. Initial cognitive testing occurred within two months of hospital discharge for nine studies (50%), with seven of those assessments being within 48 h of discharge. Twelve articles (66.7%) had at least one follow-up assessment, and eleven of them conducted at least two follow-up testing sessions, providing data for the course of cognitive changes and recovery post sepsis. See Table 2 for more information on cognitive testing.

### 3.4. Type of Measures, Congitive Domains, and Reported Scores

Seven studies (39%) only administered cognitive screening tools, with the remaining using a combination of standard neuropsychological measures. Cognitive domains typically included declarative memory, attention and working memory, processing speed, and executive function. Eleven studies (61%) used raw scores or mean raw scores, two studies (11.1%) reported percentages of scores falling 1.5 or 2 standard deviations below the mean, one study (5.6%) reported the percentage of cognitive impairment, two studies (11.1%) included t-scores (age and/or education adjusted), and two studies (11.1%) included z-scores. 

### 3.5. Statistical Analyses

Studies varied in terms of the statistical modeling employed: correlation/regression models (50%), structural equation modeling (5.6%), general estimating equations (11.1%), survival models (5.6%), parametric and nonparametric tests for group comparisons (33.3%), concordance rate models (5.6%), receiver operating characteristics (5.6%), weighted network analyses (5.6%), and linear mixed models (5.6%). Three studies (16.7%) statistically corrected for education, two studies (11.1%) reported age-adjusted t-scores, and one study (5.6%) reported education-adjusted t-scores. See Table 3 for further detail on the statistical analyses.

## 4. Discussion

Relevant research articles reviewed herein span 13 years (February 2008 to January 2021). The 18 publications addressing cognition varied with regards to the study design, demographic reporting, cognitive test type, and statistical approach. Despite 18 published articles, we continue to have a limited understanding of sepsis and cognition. Strengths include the number of studies considering cognitive domains in addition to general cognitive screeners, as well as the inclusion of an expert in cognitive assessment as part of the study team panel. Eleven studies report at least two follow-up testing sessions. Initial cognitive testing occurred within two months of hospital discharge for nine studies, with seven of those assessments being within 48 h of discharge. Across the publications, the themes suggest an acute decrease in cognitive function, with follow-up assessments showing cognitive improvement from acute levels. However, few studies considered premorbid or demographic factors within the statistical model, and less than half included a non-sepsis peer group to calculate practice effects. These design limitations challenge the accuracy of study findings, for there are associations between premorbid status and cognitive change [31,32,33,34], and there is a value to considering practice effects in post-trauma cognitive change models [35,36,37]. Our review findings suggest avenues for improvement as the field of sepsis and cognitive research moves forward.

Although more than one third of studies limited their investigation to global cognitive screening tools, the remaining studies used a combination of neuropsychological measures primarily assessing attention/working memory and declarative memory. These domains involve neurological systems typically assaulted by common comorbidities often associated with sepsis and small vessel vascular disease such as hypertension and diabetes [38,39,40]. Two studies additionally considered subdomains of measures of language (semantics) [21,23], and two looked at subdomains of higher cortical abstract reasoning [21,29], but they were limited in sample size. Hypothetically, within larger samples, comparing semantics and higher order functions to the more “vulnerable” cognitive domains would guide insight into cortical versus subcortical/white matter contributions, neuronal vulnerability, and potential mechanisms [41,42]. Longitudinal studies comparing cognitive domains in larger samples are needed.

Control groups are discussed in eight of the published investigations. Control groups used within a statistical design can provide information on “non-disease” practice effects as well as provide normative reference sources for calculating a standardized individual composite reliable change score [34,36,37,42]; the change is larger than reasonably expected due to the measurement error alone [43]. Control groups provide a reference for the expected performance. This information is particularly valuable when a researcher is concerned about the psychometric properties of cognitive measures (e.g., range of possible scores, normal distribution) and the test appropriateness for the population of interest (i.e., one would not administer a 16-word verbal list learning test to individuals with moderate cognitive impairment, for this would likely result in a poor performance and a floor effect). Patients who are acutely ill or cognitively compromised may perform at floor level; the test may be too difficult or not appropriate for identifying further deterioration. By contrast, individuals with a superior premorbid cognitive reserve who acquire sepsis may not be challenged appropriately with general cognitive screening tests. Control groups provide the research team with an external comparison to examine expected patterns of performance without complication from the disease state [37,42]. 

There was also a notable absence in the publications of the reporting of the years of education. Half of the publications reported this variable, and three considered education years in the statistical model. Given the wealth of research showing how education is a proxy for cognitive reserve, we consider the absence of educational reporting a concerning finding. The years of education have been used repeatedly in aging studies to help explain variability in cognition relative to pathology [44,45]. Cognitive reserve proxies, such as education, are shown to protect against impairment from traumatic brain injury [31,32,33] or operative/anesthesia exposure [34]. Considering education in analytical approaches may explain important variabilities in the outcome. The years of education are also a vital consideration if research teams are unable to acquire a premorbid/pre-sepsis cognitive estimation using formal test tools [46].

Sociodemographic considerations were limited, as was a consideration of medical comorbidities. Animal models [47] and some human studies [48,49] report sex as an important predictor of sepsis recovery. However, no study reported findings relative to participants’ sex. Although race or ethnicity were reported in five of the 18 studies, no study considered these variables relative to the cognitive trajectory. Today, we appreciate how race and ethnicity are surrogates for appreciating health disparities’ contribution to neurobiological responses [50]. Social determinants of health, including systemic racism experienced by many communities of color, are known to have negative ramifications on health outcomes and recovery [51,52,53]. It remains unknown if or how the cognitive sequela of sepsis differentiates across sex, race, and ethnicity. Sex, race, and ethnicity are proxies for health disparities and may modify neurobehavioral responses. This is an important area for future research. Further, individuals within different sociodemographics have a unique risk of sepsis (i.e., renal disease, congestive heart failure, myocardial infarction, chronic pulmonary disease, liver disease, peripheral vascular disease, peptic ulcer disease, and connective tissue disease) [38], and these are worthy of consideration relative to the pre-sepsis neuronal vulnerability status. We also identified a considerable patient heterogeneity in the current state of the literature with regards to the demographic characteristics and origin of sepsis. These are relevant design considerations.

Based on the current review, the authors consider sepsis cognitive research to be in an early developmental stage. There are noticeable methodological study design weaknesses that limit confidence in study findings. We encourage future researchers to consider study design suggestions from the fields of epilepsy (see [54] for a review of neuropsychology and epilepsy) and perioperative cognitive research—both of which have aggressively addressed challenges in longitudinal cognitive study design problems since the 1990s [42]. While sepsis poses a unique challenge due to the unexpected nature of the illness, researchers may wish to consider the benefits of including a control group, premorbid intellectual estimates, education, and other sociodemographic factors in future investigations. It should be noted that our review spanned a considerable timeframe (from February 2008 to January 2021), where the definition and management of sepsis rapidly evolved. We encourage a repeated literature review over the next 5 years.

## Figures and Tables

**Figure 1 jcm-10-04269-f001:**
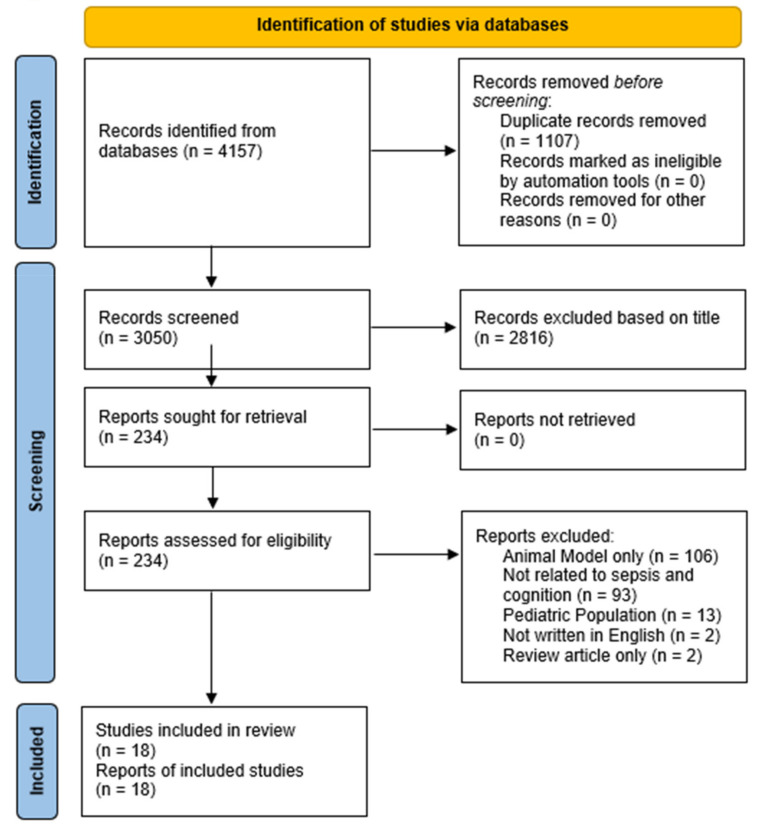
Publication Identification Process.

**Table 1 jcm-10-04269-t001:** Demographic characteristics of the retained studies.

1st Author	Year	Sepsis Sample (n)	Control Sample (n)	Neuro Expert	Mean Age	Education Reported	Sex (male:female)	Race/Ethnicity
Regazzoni [13]	2008	137	No	No	80.81	NR	67:70	NR
Girard [14]	2010	77	No	Yes	61	Median = 12 (IQR = 10–13) years	40:37	NR
Iwashyna [15]	2010	623	No	No	76.9	NR	281:362	Black: 128 Hispanic: 44 White: 451
Davydow [16]	2012	517	No	No	76.1	≤HS = 38.5%; Some College = 34.8; >College degree = 26.7%	235:282	White: 416 Black: 95 Other: 6
Merli [17]	2013	31	Yes (23)	No	NR	NR	NR	NR
Semmler [18]	2013	25	Yes (19)	Yes	55.64	NR	13:12	NR
Götz [19]	2015	36	Yes (30)	Yes	59.8	NR	24:12	NR
Götz [20]	2016	36	Yes (30)	Yes	59.8	NR	24:12	NR
Needham [21]	2016	83	Yes (106)	Yes	52	Mean = 13.9 ± 2.2 years	39:44	White: 68 Non-White: 15
Pierrakos [22]	2017	28	No	Yes	67.3	NR	NR	NR
Brown [23]	2018	40	No	Yes	NR	<HS = 13%; HS = 20%; Associate degree = 18%; Higher education = 10%	19:21	NR
Calasavara [24]	2018	33	No	Yes	49	Mean = 7 (IQR = 4–8) years	14:19	NR
Kang [25]	2018	36	No	No	67.8	Mean = 7.4 ± 5.8 years	29:7	NR
Orhun [26]	2019	MMSE < 24 = 7 MMSE 24–30 = 14	Yes (33)	Yes	MMSE < 24: 57.3 ± 3.1 MMSE 24–30: 53.2 ± 3	MMSE < 24: Mean = 7.1 ± 1.1 years MMSE 24–30: Mean = 8.7 ± 1.0 years	MMSE < 24 = 4:3 MMSE 24–30 = 8:6	NR
Seidel [27]	2019	20	Yes (44)	Yes	53.8	NR	9:11	NR
Mankowski [28]	2020	328	No	No	Young group = 35 Middle group = 58 Old group = 72	NR	176:152	White: 293 African American: 30 American Indian: 1 Other: 1 Unknown: 1
Brown [29]	2021	30	No	Yes	56	<HS = 10%; HS = 23%; Some College = 33%; Associate degree = 7%; Bachelor degree = 17%; >Bachelor degree = 10%	13:17	Asian: 1 Native Hawaiian/other pacific islander: 1 White: 28
Wang [30]	2021	840	Yes (20,893)	No	64.3	<HS = 12.7%; HS = 26.8%; Some College = 29.3%; >College = 31.2%	NR in the final sample	NR in the final sample

Abbreviations: HS = High School; IQR = Interquartile Range; MMSE = Mini-Mental State Exam; NR = Not Reported.

**Table 2 jcm-10-04269-t002:** Study designs and cognitive testing measures.

1st Author	Year	Baseline Testing	Time of Testing	Tests (Estimated Length)	Cognitive Domains	Reported Score
Regazzoni [13]	2008	No	Admit	MMSE (10’)	Global	Raw score
Girard [14]	2010	No	3 months, 12 months post sepsis	MMSE, WAIS-III DS, TMT A&B, Coding, RAVLT, RCFT (Copy & Delay), VF (70’)	Global, ATT, DM, PS, VC, WM	T-scores (age and education adjusted based on specific population norms)
Iwashyna [15]	2010	Yes	1998—death or 2006	M-TICS (10’)	Global	% of cognitive impairment in sample
Davydow [16]	2012	Yes	Mean = 7y post sepsis	TICS or TICS-27 (10’)	Global	TICS raw score
Merli [17]	2013	No	Admit, 3 months post discharge	MMSE, TMT A & B, Digit Symbol (18’)	Global, ATT, PS, WM	Z-Scores (test specific population norms)
Semmler [18]	2013	No	6–24 months post discharge	German Vocab. Test, NeuroCogFx, TMT A&B, AVLT, RCFT (70’)	ATT, DM, FM, PI, STM, VF	Mean unweighted score, (zDiff) = ((Cognitive Test z-score) − (Multiple Choice Word Test-B z-score))) (test specific population norms)
Götz [19]	2015	No	0–2 months, 5–8 months, 10–15 months post ICU discharge	DemTect & CDT (15’)	Global	DemTect raw score
Götz [20]	2016	No	0–2 months, 5–8 months, 10–15 months post ICU discharge	DemTect & CDT (15’)	Global	DemTect raw score
Needham [21]	2016	No	6 months and 12 months post ICU discharge	Hayling Sentence Completion Test, VF, WAIS-III Similarities & DS, WMS-III LM (50’)	ATT, DM, EF, VR, WM	Percentage of scores 1.5 SD below the mean
Pierrakos [22]	2017	No	Sepsis discharge (~8d post sepsis onset)	MMSE, CDT (15’)	Global	MMSE raw score & MMSE recall sub-score
Brown [23]	2018	No	Sepsis discharge, 3 months, 6 months	MoCA, VF, WAIS-IV DS Similarities, WMS-IV LM, Hayling Sentence Completion Test (70’)	Global, ATT, DM, EF, VF, VR, WM	Percentage of scores 1.5 SD and 2.0 SD below the mean (test specific population norms)
Calasavara [24]	2018	No	24 h post discharge, 1 year (median 393 days)	MMSE, CERAD (Verbal Fluency, TMT A&B, BNT, List Learning, Praxis, List Recall & Recognition, Praxis Recall) (70’)	Global, DM, EF, Language (naming, comprehension), PS, VF	MMSE raw score, CERAD mean scores
Kang [25]	2018	No	48 hours after ICU admit	K-MoCA; K-MMSE; CoSAS-S (30’)	Global	Raw scores
Orhun [26]	2019	No	0, 3 months, 12 months post sepsis	MMSE & ACE-R (Orientation-attention, Memory, VF, Language, Visuospatial Function) (35’)	Global	MMSE raw scores; ACE-R Sub-scores means
Seidel [27]	2019	No	2.6 ± 1.9 years post sepsis	TAP, Go-No-Go paradigm, German version of AVLT (40’)	ATT, DM, WM	T-scores (age-adjusted based on test specific population norms)
Mankowski [28]	2020	No	3, 6, 12 months post discharge	MMSE, HVLT-R (Total recall, Delayed recall, Retention), COWA (30’)	Global, DM, VF	Mean raw scores
Brown [29]	2021	No	6 months post discharge	Hayling Sentence Completion Test (5’)	EF	Mean raw scores
Wang [30]	2021	Yes	6 months post sepsis; every 2 years (2006–2017)	SIS, CERAD: WLL, WLD, AFT (20’)	Global	Mean raw scores

Abbreviations: ACE-R = Addenbrooke’s Cognitive Examination Revised; AFT = Animal Fluency Test, ATT = Attention; AVLT = Auditory Verbal Learning Test; BNL = Below Normal Limits; BNT = Boston Naming Test; CERAD = Consortium to Establish a Registry for Alzheimer’s Disease, CoSAS-S = Computer Cognitive Senior Assessment System-Screen; COWA = Controlled Oral Word Association; DM = Declarative Memory; DS = Digit Span; EF = Executive Functioning; FM = Figural Memory; HS = High School; HVLT-R = Hopkins Verbal Learning Test - Revised; K-MMSE = Korean Mini Mental State Exam; K-MoCA = Korean-Montreal – Cognitive Assessment; MMSE = Mini Mental State Exam; MoCA = Montreal – Cognitive Assessment; NP = Neuropsychological; NR = Not Reported; PI = Premorbid Intelligence; RAVLT = Rey Auditory Verbal Learning Test; RCFT = Rey-Osterreith Complex Figure; SD = Standard Deviation; SIS = Six-Item Screener; STM = Short Term Memory; TAP = Test of Attentional Performance; TICS-27 = Telephone Interview for Cognitive Status-Modified; Six-Item Screener = TICS-M = Telephone Interview for Cognitive Status-Modified; TMT = Trail Making Test; VF = Verbal Fluency Test; VR = Verbal Reasoning; WM = Working Memory; WAIS-III = Weschler Adult Intelligence Scale Third Edition; WAIS-IV = Weschler Adult Intelligence Scale Fourth Edition; WLD = Word List Delayed Recall; WLL = Word List Learning; WM = Working Memory; WMS-III LM = Weschler Memory Scale Third Edition Logical Memory; WMS-IV LM = Weschler Memory Scale Fourth Edition Logical Memory.

**Table 3 jcm-10-04269-t003:** Statistical Analyses and Corrections, and Summary of Cognitive Findings.

1st Author	Year	Statistical Method	Education Correction	Sex and/or Race Correction	Findings
Regazzoni [13]	2008	Cox proportional hazard model and Kaplan-Meier test	No	No	Sepsis survivor MMSE mean = 20.14; scores predicted 1-year mortality.
Girard [14]	2010	Multiple nonlinear regression, propensity score matching	Yes	No	Cognitive impairment at 3 months: 79% (62% severe); 12 months: 79% (36% severe). As duration of delirium increased, cognition decreased.
Iwashyna [15]	2010	Latent growth curve models, random effects models, and logistic regressions.	No	No	Rate of moderate or severe cognitive impairment among survivors (pre-sepsis) increased from 6.1% (95% CI: 4.2%, 8.0%) before severe sepsis to 16.7% (95% CI: 13.8%, 19.7%) at the 1st assessment after severe sepsis.
Davydow [16]	2012	Paired *t*-test, Pearson’s correlation, logistic regression.	No	No	Cognitive impairment in 18% of severe sepsis survivors. Pre-sepsis depression was the greatest predictor of post-sepsis cognitive impairment.
Merli [17]	2013	Logistic regression.	Yes	No	No sepsis survivors were cognitively impaired, but 42% of sepsis survivors showed a decline in performance.
Semmler [18]	2013	Student t-tests, Pearson’s correlation, ANOVA, and MANCOVA.	No; Estimated premorbid verbal abilities instead.	No	Sepsis survivors impaired in 8 of 9 subtests (mainly learning and memory). Non-septic ICU survivors showed deficits in 6 subtests.
Götz [19]	2015	General estimating equations.	No	No	Sepsis survivors were impaired on periodic visual stimulation (familiar and unfamiliar pictures) and scored lower than HCs on the DemTect and CDT at all time points.
Götz [20]	2016	General estimating equations.	No	No	Sepsis survivors scored lower than HCs on the DemTect and CDT at all time points.
Needham [21]	2016	Joint survival models, linear regressions and logistic random intercept regression models.	No	No	38% at 6m post-sepsis had cognitive impairment; 28% at 12m.
Pierrakos [22]	2016	Multivariable linear regression.	If education >12 years, no MMSE was given; CDT only.	No	50% of sepsis survivors had cognitive decline, with greatest decline in information recall.
Brown [23]	2018	Concordance correlation coefficient and Fisher’s exact tests.	No	No	At discharge, 90% of survivors had MoCA< 25 (BNL). At 3 months, 70% BNL; 6 months, 57% BNL. Neuropsychology: 3 months, 43% impaired; 6 months, 57% impaired. WAIS-IV DS was the one test performance that did not improve from 3 to 6 months.
Calasavara [24]	2018	Student *t*-test, Mann-Whitney U, chi-square test, one sample *t*-test, marginal models, stepwise regression.	No	No	At discharge, survivors had lower MMSE and poor constructional praxis (*p* < 0.001). At 1 year, all performances normalized except for the BNT (*p* = 0.193) and constructional praxis (*p* < 0.001).
Kang [25]	2018	Receiver Operating Characteristics.	No	No	53.1% of sepsis survivors were cognitively impaired on MMSE; 65.6% of sepsis survivors were cognitively impaired on MoCA
Orhun [26]	2019	Mann-Whitney U or Kruskal-Wallis tests, Dunn’s post-hoc test, Spearman correlation.	No	No	Initial mean MMSE = 25.4 ± 3.9; 3 months: 27.8 ± 2.8; 12 months: 28.4 ± 1.4
Seidel [27]	2019	Two-tailed student *t*-test, Pearson correlation.	No	No	55% of survivors had deficits in 1–2 domains and 20% in 3 or more domains. Primary difficulties in learning, alertness, working memory, and memory decay rate.
Mankowski [28]	2020	Fisher exact test and Kruskal-Wallis test.	No	No	Young adults performed better than middle-aged and older adults. No group differences between the middle-aged and older adults.
Brown [29]	2020	Weighted network analysis.	No	No	20% of survivors had impairment in executive domain according to the Hayling Sentence Completion Test
Wang [30]	2021	Multivariable linear mixed-effects models.	Yes	No	SIS scores of sepsis survivors improved. AFT scores decreased, while WLD and WLL scores increased.

Abbreviations: AFT = Animal Fluency Test, ANOVA = Analysis of Variance; BNL = below normal limits; CDT = clock drawing test; CI = confidence interval; HC = Healthy Controls; ICU = Intensive Care Unit; MANCOVA = Multiple Analysis of Covariance; MoCA = Montreal Cognitive Assessment; MMSE = Mini Mental State Examination; SD = Standard Deviation; SIS = Six-Item Screener; WLD = Word List Delayed Recall; WLL = Word List Learning.

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
