# Peer review of "Sepsis and Cognitive Assessment"

_jcm, 2021, doi:10.3390/jcm10184269_

Round 1
Reviewer 1 Report
This is a very nice review of an important deficiency in current research. Having more objective and better designed studies in future will help us get better data points which in turn will guide us in management of sepsis in the future. Also we will have a better assessment of the morbidity associated with sepsis.
In figure 1, I would suggest a little more clarification: Initial records identified are listed as n=1 only, which may be an error. Also I would suggest reason for exclusion of the 2816 records to be listed.
Author Response
Thank you for your constructive and insightful review.
We agree that with time and more data points sepsis management will trend toward optimization. Below are our responses to your suggestions:
- The n=1 was an error and has been rectified. See Figure 1.
- Further clarification was provided in Figure 1 to explain that 2,816 records were excluded based upon title ineligibility (i.e. not relating to sepsis or cognition).
Reviewer 2 Report
Dear authors,
Congratulations for this great review, the work is relevant and analyze a field of research in sepsis little known worlwide but so important for the long term outcomes of survivors. The aims, review methodology and Flow of analysis of publications revised are well explained and the results are well described in the tables and text. Discussion is very interesting and I agree with the conclusions and the need of large and methodologically well defined future studies in this research area of sepsis.
I suggest minor changes for this review article,
The literature review process is well explained and works included in the final analysis are relevant. The timeline of the publications comprises last 13 years (2008 to 2021) and definition, research, management and outcomes in sepsis have changed and improved along this large period. I suggest including this concerning issues about heterogeneity in patients inclusion and results in the discussion.
In the Introduction section (lines 27-28) you defined sepsis in a generally manner "as tissue and organ damage resulting form the host reponse to infection", I think It´s more precise to reference this definition to the last actualization in sepsis definition "Sepsis-3 definition" by Singer et al published in JAMA "Sepsis should be defined as life-threatening organ dysfunction caused by a dysregulated host response to infection" and add this to the reference list (Singer M, et al. The Third International Consensus Definitions for Sepsis and Septic Shock (Sepsis-3) JAMA. 2016 Feb 23; 315(8): 801–810. )
Author Response
Thank you for pointing the factor of heterogeneity out. We added a few sentences in different paragraphs of the discussion section to address the heterogeneity of the samples reviewed and the considerable timeframe of the review. Changes are reflected on page 5 and 6.
Regarding the updated Sepsis-3 definition, we revised the introductory paragraph on page 1 to reference the latest definition of sepsis by Singer and colleagues. This paper has been added to the reference list.